# Evaluation of Bicanalicular Nasolacrimal Duct Intubation as an Adjunct in Surgical Ectropion Correction

**DOI:** 10.3390/medicina58081051

**Published:** 2022-08-04

**Authors:** Anthia Papazoglou, Triantafyllia Chrysochoou, David Goldblum, Markus Tschopp, Tim J. Enz

**Affiliations:** 1Department of Ophthalmology, Cantonal Hospital Aarau, 5000 Aarau, Switzerland; 2Department of Ophthalmology, University Hospital Basel, 4056 Basel, Switzerland; 3Department of Ophthalmology, Pallas Kliniken, 4600 Olten, Switzerland; 4Department of Ophthalmology, University Hospital of Bern, University of Bern, 3010 Bern, Switzerland

**Keywords:** entropion, entropion surgery, recurrence rate, nasolacrimal duct intubation

## Abstract

*Background and Objectives*: We aimed to analyze and compare the outcomes of conventional ectropion surgery procedures with and without concurrent bicanalicular nasolacrimal duct intubation to identify if the combination of procedures could serve as a novel surgical approach to treat lower eyelid ectropion. *Materials and Methods*: A retrospective review of all patients who underwent surgical correction for lower eyelid ectropion at the Cantonal Hospital of Aarau between January 2019 and December 2020 was performed. Patient medical records were examined for etiology, surgical correction technique and intra- and postoperative complications. The postoperative punctal position, the pre- and postoperative epiphora and reoperation rate were also documented. Two study groups consisting of cases with isolated and combined procedures were compared, with respect to postoperative punctual and lower lid position. *Results*: A total of 53 lower eyelids (35 patients) were included in this study. Six months postoperatively, the correct punctum position (*p* = 0.1188) and improvement of epiphora (*p* = 0.7739) did not significantly differ between the two groups. More complications were seen in the nasolacrimal duct intubation group (*p* = 0.0041), which consisted of cheese wiring and one tube dislocation. *Conclusion*: In our study, bicanalicular nasolacrimal intubation during ectropion surgery does not seem to improve the outcome of ectropion surgery and is, therefore, not recommended on a routine basis.

## 1. Introduction

Lower eyelid ectropion is one of the most frequent eyelid malpositions [1]. It is characterized by the eversion of the lower lid margin, leading from mild epiphora and dry eye disease to severe exposure keratopathy [1,2]. The etiology of lower eyelid ectropion varies; the most common etiologies are age-related involutional changes (dehiscence of the anterior and posterior lamellas and degeneration of the orbicularis muscle), leading to increased laxity of the canthi [1]. Other causes, such as congenital, paralytic, cicatricial and mechanical ectropion, are less frequent.

Several surgical procedures for ectropion correction have been described, for example, lateral tarsal strip, lateral canthoplasty/canthopexy, lid shortening through a full-thickness wedge excision, combined with or without medial spindle/spiraling suture to address vertical eyelid laxity and correct punctal eversion [1,2,3,4,5,6,7,8]. Larger surgical interventions involving skin grafting or scar correction are sometimes needed for cicatricial and mechanical ectropions [5]. Favorable safety profiles and high success rates are inherent to all the above-established surgical techniques; however, ectropion occasionally reoccurs after all types of ectropion correction surgery.

Mono- or bicanalicular nasolacrimal duct intubation is conventionally used to address congenital or acquired nasolacrimal duct obstructions or for traumatic nasolacrimal duct lacerations [9,10,11]. In the latter, the tube serves to readapt the lacrimal duct and prevent its scarring, yet it may also function to stabilize the lacerated lid in the correct position [12]. A case report study conducted by Higaki et al. published in 2013 described nasolacrimal duct obstruction treated with canaliculoplasty and nasolacrimal duct intubation, leading to a resolution of a concurrently present medial lower eyelid ectropion [12]. The authors suggested that the constant forces exhibited onto the lower eyelid, due to the intubation stabilizing the lower eyelid in its correct position. However, they did not examine a cohort of patients. Inspired from their suggestion, we hypothesized that nasolacrimal duct intubation might be useful for stabilizing the lower eyelid also in cases of surgical ectropion repair as well. 

Since epiphora is often caused by a combination of partial nasolacrimal duct obstructions and lower eyelid laxity nasolacrimal duct intubation is often combined with conventional surgical procedures for ectropion correction. In our study, we report our experience with nasolacrimal duct intubation as an adjunct to conventional lid shortening or punctal inversion procedures for lower eyelid ectropion correction. We aimed to analyze and compare the postoperative lower punctal positioning after ectropion surgery procedures with or without bicanalicular intubation. To our knowledge, nasolacrimal duct intubation has not been investigated as an adjunct to conventional ectropion correction procedures in any other cohort studies before. 

## 2. Methods

### 2.1. Study Design and Ethics

This study was a retrospective case series. It was conducted in accordance with the Declaration of Helsinki and approved by the competent ethics committee (Ethikkommission Nordwest- und Zentralschweiz EKNZ, approval identification number 2021-00011).

### 2.2. Surgical Procedure

All patients underwent lower eyelid ectropion correction surgery. The surgical procedures used in our cohort study either aimed to correct the horizontal laxity or the horizontal and the vertical laxity of causing the ectropion. The surgical approaches, aiming to restore the horizontal laxity, used were lateral tarsal strip, lateral canthopexy, partial lateral tarsorrhaphy and KZ-Procedure, while the ones aiming to restore horizontal and the vertical laxity were Lazy-T and any surgery combined with a diamond-shaped medial spindle/spiraling suture for punctal inversion. In one cicatricial ectropion case, in which skin grafting was needed, a Tripier’s flap was also performed.

For nasolacrimal duct intubation, a commercially available bicanalicular silicone self-pushed tube was used (Nunchaku©, FCI, Pembroke, MA, USA) (Figure 1). Both puncta were widened intraoperatively using a punctum dilator. Then, the canaliculi were probed and the lacrimal system was irrigated with 0.9% sodium chloride solution. Finally, the ends of the Nunchaku tube were inserted via the lower and the upper punctum, and the nasolacrimal duct was intubated in its entire length.

All patients were examined within the first three postoperative days to ensure the correct tube positioning. The tube was then left in place for approximately three months and removed during a postoperative outpatient follow-up consultation. The final follow-up examination was conducted approximately six months postoperatively, at which point clinical outcomes were recorded.

### 2.3. Subjects and Retrospective Patients’ Records Reviewing

All theater lists at our institution were manually screened for cases of ectropion repair performed between January 2019 and December 2020. Patient medical records (pre- and postoperative photographs, surgical logs, consultation notes) were investigated with respect to baseline patient characteristics, coexisting partial lacrimal obstruction, the etiology of the ectropion, the localization of the lower eyelid laxity, surgical techniques, intra- and postoperative complications, the postoperative punctal position, epiphora and reoperation rate. Epiphora was graded according to the Munk Scale and dry eye disease according to the Oxford grading scheme [13,14].

The ectropion cases were divided into two groups: Group 1 included cases with ectropion surgery without nasolacrimal duct intubation, and group 2 included cases with ectropion surgery with bicanalicular nasolacrimal duct intubation. For the bicanalicular nasolacrimal duct intubation, patients were chosen with concurrent partial lacrimal duct obstructions.

### 2.4. Data Analysis

Statistical analysis was performed with the Prism 5.0c (GraphPad Software, La Jolla, CA, USA). Descriptive statistics included medians and ranges. Nonparametric tests were performed for all values not normally distributed. The Mann–Whitney test or the Fisher’s exact test in case of dichotomous datasets were performed to compare the two groups. The level of significance was set at a *p* value of <0.05.

## 3. Results

In total, 35 patients (53 lower eyelids) were included in this study. Group 1 comprised 18 patients (27 lower eyelids) who underwent isolated lower eyelid ectropion, and group 2 consisted of 17 patients (26 lower eyelids) who underwent lower eyelid ectropion. Patients in group 2 had a small degree of nasolacrimal stenosis. Patient characteristics and the results of the statistical analysis are shown in Table 1.

Preoperatively, patients in group 2 had a correct position of the punctum (19.2%) less frequently than patients in group 1 (44.4%). Six months postoperatively, both group 2 (65.4%) and group 1 (85.2%) usually had a correct punctual position. The improvement rate did not significantly differ (*p* = 0.4837) between group 1 (71.4%) and group 2 (57.1%).

The ectropion recurrence rate was low in both groups. In group 1, one eyelid had to be reoperated at two months postoperatively due to suture breakage. In group 2, two patients had to undergo a revision surgery due to suture breakage in one patient and ectropion recurrence in the other patient. The latter patient had a cicatricial ectropion, known for its higher recurrence rate.

Other postoperative complications related to the bicanalicular nasolacrimal duct intubation included a linear dilatation of the tear duct punctum in the sense of cheese wiring and one dislocation of the tube. The total number of postoperative complications was significantly higher in group 2 than group 1 (*p* = 0.0041) due to the relatively high rate of cheese wiring. No intraoperative complications were described.

The median preoperative epiphora according to the Munk Scale did not significantly differ between the two groups (*p* = 0.4466). The postoperative Munk Scale improved similarly in both groups (*p* = 0.7739).

The rate of preoperative dry eye disease was higher in group 2 than group 1 (*p* = 0.0108), and this difference appeared to be even bigger postoperatively (*p* < 0.0001). This suggests that the bicanalicular nasolacrimal duct intubation might have contributed to a better tear drainage.

Since the main outcome of our study was the stabilization of the lower eyelid expressed through the punctual positioning, we examined the postoperative outcomes according to type of the surgical approach used. In group 2, the quota between surgical procedures aiming to correct only horizontal laxity versus horizontal and vertical laxity was 13/26 (50%). The equivalent quota in group 1 was 3/27 (11.1%). This difference was statistically significant (*p* = 0.028). The percentage of eyelids with improved postoperative punctum position was 44.4% (group 2) or 63.6% (group 1) after a surgical procedure addressing only the horizontal laxity (*p* = 0.6534) and 66.7% (group 2) or 100% (group 1) after a surgical procedure addressing horizontal and vertical laxity (*p* = 0.5165).

## 4. Discussion

To our knowledge, this cohort study is the first attempt to investigate the benefits and hazards of bicanalicular nasolacrimal intubation as an adjunct to ectropion surgery. In our study, we could not verify the hypothesis that the bicanalicular nasolacrimal duct intubation may lead to better punctum positioning or better stabilization of the lower eyelid. Furthermore, the bicanalicular nasolacrimal duct intubation leads to more postoperative complications. The only clear benefit of the bicanalicular nasolacrimal duct intubation observed in our study was a reduction in epiphora. This is, however, expected as the tube dilated and reopened the partially stenosized lacrimal duct. 

We acknowledge some limitations of our study. First, the size of the groups were relatively small. Second, the groups were not completely homogenous in terms of patient age and the surgical approaches used. However, the overall result of all surgical techniques used in our hospital is the horizontal eyelid shortening and, therefore, the technique to achieve this should not make a clinically relevant difference. Another parameter that differentiates the groups from each other is that group 2 patients had two reasons for epiphora: nasolacrimal duct stenosis and ectropion, while the group 1 patients had a patent lacrimal system. Nevertheless, this difference should not influence the main outcome of our study, which was the postoperative stabilization of the lower eyelid expressed through the punctual positioning. 

In conclusion, the results of our study suggest that the default nasolacrimal intubation is not recommended in ectropion surgery.

## Figures and Tables

**Figure 1 medicina-58-01051-f001:**
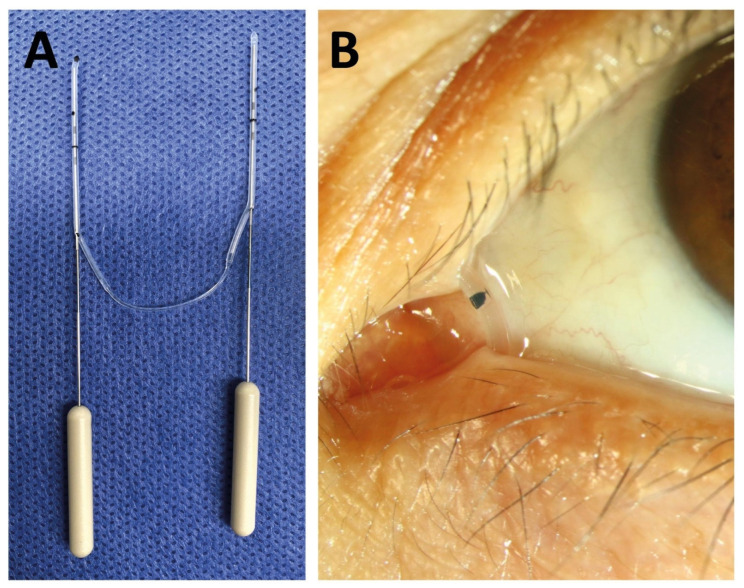
Picture of the Nunchaku tube before implantation (**A**) and when implanted (**B**).

**Table 1 medicina-58-01051-t001:** Patients’ characteristics, pre- and postoperative findings. Epiphora was assessed with the Munk Scale (0: No epiphora; 1: Epiphora requiring dabbing less than twice per day; 2: Epiphora requiring dabbing 2–4 times per day; 3: Epiphora requiring dabbing 5–10 times per day; 4: Epiphora requiring dabbing more than 10 times per day; 5: Constant epiphora). Dry eye severity was assessed according to the Oxford grading scheme.

	Total	Group 1	Group 2	Significance Group 1 vs. Group 2
**Patients enrolled,** number **Patients,** eyes	35 53	18 (51.4%) 27 (51%)	17 (48.6%) 26 (49%)	
- male - age (years), median (range)	37 (70%) 76 (57–83)	20 (74%) 71 (57–93)	17 (65%) 78.5 (61–86)	*p* = 0.0218
**Observation time (days)**, median (range)	286 (320)	298 (320)	256 (227)	*p* = 0.0656
**Operated side**				
- right eye	26	12	14	
- left eye	27	15	12	
**Surgeries aiming to restore vertical laxity**	16	3	13	*p* = 0.028
**Epiphora preoperative,** median (range) 0 No epiphora 1 2 3 4 5 Constant epiphora	3 (0–5) 5 9 2 15 6 16	3 (0–5) 5 6 0 5 2 8	3 (1–5) 0 3 2 10 4 8	*p* = 0.4466
**Epiphora postoperative,** median (range) 0 No epiphora 1 2 3 4 5 Constant epiphora	1 (0–5) 26 11 6 9 0 1	0 (0–5) 17 6 0 4 0 1	1 (0–3) 9 5 6 5 0 0	*p* = 0.1481
**Epiphora improvement**, median (range)	2 (4)	1 (4)	2 (3)	*p* = 0.7739
**Dry eye severity preoperative,** median (range) 0 Absent 1 Minimal 2 Mild 3 Moderate 4 Marked	1 (0–3) 11 17 15 10 0	1 (0–3) 9 10 4 5 0	2 (0–3) 2 7 11 5 0	
**Dry eye severity postoperative,** median(range) 0 Absent 1 Minimal 2 Mild 3 Moderate 4 Marked	1 (0–3) 15 17 17 4 0	0 (0–3) 15 6 5 2 0	2 (1–3) 0 11 12 2 0	*p* < 0.0001
**Preoperative correct position of punctum**	17	12 (44.4%)	5 (19.2%)	*p* = 0.0772
**Postoperative correct position of punctum**	40	23 (85.2%)	17 (65.4%)	*p* = 0.1188
**Improved position of punctum**	22	10/14 (71.4%)	12/21 (57.1%)	*p* = 0.4875
**In eyes aiming to restore vertical laxity**	11	3/3 (100%)	8/12 (66.7%)	*p* = 0.5165
**In other eyes**	11	7/11 (63.6%)	4/9 (44.4%)	*p* = 0.6534
**Total Complications** - Suture granuloma - Secondary surgery due to persisting ectropion - Secondary surgery due to wound dehiscence - Cheese wiring - Nunchaku dislocation		2 (7.4%) 1 0 1 0 0	11 (42.3%) 0 1 1 8 1	*p* = 0.0041

## Data Availability

Data supporting the findings of this study are available within the article.

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
