# Peer review of "Evaluation of Bicanalicular Nasolacrimal Duct Intubation as an Adjunct in Surgical Ectropion Correction"

_medicina, 2022, doi:10.3390/medicina58081051_

Round 1

Reviewer 1 Report

Interesting view-point for checking lacrimal intubation's impact on corrective procedures long term. What made you consider this approach? Its not clear why this is required? Perhaps add a paragraph for reasoning please. 

Kindly ensure a grammar review and spelling review. There has been interchangeable use of ectropium vs ectropion - which should be clarified and unified throughout the script. 

Author Response

Dear reviewer,

Thank you for your reviewing our manuscript and the valuable comments on our manuscript. Please find our responses one by one as followings. If there is anything else we can do to improve the quality of the manuscript or if you have any questions or concerns, do not hesitate to contact us.

Interesting view-point for checking lacrimal intubation's impact on corrective procedures long term. What made you consider this approach? Its not clear why this is required? Perhaps add a paragraph for reasoning please. 

We now have added the following paragraph to the introduction section that explains why we considered and studied this approach:

"A case report study conducted by Higaki et al published in 2013 described nasolacrimal duct obstruction treated with canaliculoplasty and nasolacrimal duct intubation, leading to a resolution of a concurrently present medial lower eyelid ectropion.[12] The authors suggested that the constant forces exhibited onto the lower eyelid due to  the intubation stabilized the lower eyelid in its correct position. However, they did not examine a cohort of patients. Inspired from their suggestion, we hypothesized that nasolacrimal duct intubation might be useful for stabilizing the lower eyelid also in cases of surgical ectropion repair as well."

Kindly ensure a grammar review and spelling review. There has been interchangeable use of ectropium vs ectropion - which should be clarified and unified throughout the script. 

Thank you for pointing out this mistake. The manuscript has now been reviewed by a medical writer.

Reviewer 2 Report

The paper is written clearly and interesting. It might be interesting to compare also the outcomes depending on the surgical procedure, percentage of used procedure and its influence to postoperative position of punctum. I would love to see that. You shall also revise some spelling mistakes ( just a few).

Author Response

Dear reviewer,

Thank you for your reviewing our manuscript and the valuable comments on our manuscript. Please find our responses one by one as followings. If there is anything else we can do to improve the quality of the manuscript or if you have any questions or concerns, do not hesitate to contact us.

The paper is written clearly and interesting. It might be interesting to compare also the outcomes depending on the surgical procedure, percentage of used procedure and its influence to postoperative position of punctum. I would love to see that. You shall also revise some spelling mistakes ( just a few).

We agree that it would be interesting to see the outcomes depending on the surgical procedure. However, as the number of each procedure probably would be too small for useful statistical analysis, we have now grouped the patients in a group that aimed to restore only the horizontal lid laxity, and a group that aimed to restore the horizontal and vertical lid laxity. In addition, we could offer a supplementary table with all patients and their surgical procedure.

The manuscript now has also been reviewed by a medical writer.

We have added the following paragraph to the methods section:

"All patients underwent lower eyelid ectropion correction surgery. The surgical procedures used in our cohort study either aimed to correct the horizontal laxity or the horizontal and the vertical laxity of causing the ectropion. The surgical approaches aiming to restore the horizontal laxity used were lateral tarsal strip, lateral canthopexy, partial lateral tarsorrhaphy, KZ-Procedure, while the ones aiming to restore horizontal and the vertical laxity were Lazy-T and any surgery combined with a diamond-shaped medial spindle/spiraling suture for punctal invertion. In one cicatricial ectropion case, in which skin grafting was needed, a Tripier’s flap was also performed."

We have added the following paragraph to the results section and listed these two groups in table 1:

"Since the main outcome of our study was the stabilization of the lower eyelid expressed through the punctual positioning we examined the postoperative outcomes according to type of the surgical approach used. In group 2 the quota between surgical procedures aiming to correct only horizontal laxity versus horizontal and vertical laxity was 13/26 (50%). The equivalent quota in group 1 was 3/27 (11.1%). This difference was statistically significant (p=0.028). The percentage of eyelids with improved postoperative punctum position was 44.4% (group 2) or 63.6% (group 1) after a surgical procedure addressing only the horizontal laxity (p=0.6534) and 66.7% (group 2) or 100% (group 1) after a surgical procedure addressing horizontal and vertical laxity (p=0.5165)."

Improved position of punctum

22

10/14 (71.4%)

12/21 (57.1%)

p=0.4875

    In eyes aiming to restore vertical laxity

11

3/3 (100%)

8/12 (66.7%)

p=0.5165

    In other eyes

11

7/11 (63.6%)

4/9 (44.4%)

p=0.6534

Reviewer 3 Report

I read the paper entitled Could routine bicanalicular tear duct intubation improve surgical ectropion correction?very carefully. The topic of the article is interesting. The authors evaluated the possible role of nasolacrimal intubation combined with ectropion surgery as it was described in one case some years ago.  Some minor corrections must be made by the authors.

The Title should be more clearly without question at the end. I suggest: Evaluation of nasolacrimal duct intubation on improvement for surgical ectropion correction.

In Methods the authors correctly described the type of nasolacrimal tube (Nunchaku tube). But in the Results: it will be better to used two groups - first group: ectropion surgery without nasolacrimal duct tube intubation and second group: ectropion surgery with nasolacrimal duct intubation without the use of the tube name.

In Table 1 - I suggest to use the same group classification – group 1: ectropion surgery without nasolacrimal duct tube intubation and group 2: ectropion surgery with nasolacrimal duct intubation and not use the name of the tube.

In Discussion the authors should emphasize that the results of other nasolacrimal tube could be the same.

In Conclusion, authors find out that additional tube implantation together with ectropion surgery did not contribute to better results.

Author Response

Dear reviewer,

Thank you for your reviewing our manuscript and the valuable comments on our manuscript. Please find our responses one by one as followings. If there is anything else we can do to improve the quality of the manuscript or if you have any questions or concerns, do not hesitate to contact us.

I read the paper entitled “Could routine bicanalicular tear duct intubation improve surgical ectropion correction?” very carefully. The topic of the article is interesting. The authors evaluated the possible role of nasolacrimal intubation combined with ectropion surgery as it was described in one case some years ago.  Some minor corrections must be made by the authors.

The Title should be more clearly without question at the end. I suggest: Evaluation of nasolacrimal duct intubation on improvement for surgical ectropion correction.

Thank you for this suggestion. We now have changed the title to:

" Evaluation of bicanalicular nasolacrimal duct intubation as an adjunct in surgical ectropion correction."

In Methods the authors correctly described the type of nasolacrimal tube (Nunchaku tube). But in the Results: it will be better to used two groups - first group: ectropion surgery without nasolacrimal duct tube intubation and second group: ectropion surgery with nasolacrimal duct intubation without the use of the tube name.

In Table 1 - I suggest to use the same group classification – group 1: ectropion surgery without nasolacrimal duct tube intubation and group 2: ectropion surgery with nasolacrimal duct intubation and not use the name of the tube.

Thank you for this suggestion. This now been changed accordingly throughout the manuscript.

In Discussion the authors should emphasize that the results of other nasolacrimal tube could be the same.

We now replaced "Nunchaku tube" throughout the manuscript with " bicanalicular nasolacrimal duct intubation", with the exception of the methods section. And have changed the conclusion to:

"In conclusion, the results of our study suggest that the default nasolacrimal intubation is not recommended in ectropion surgery."

In Conclusion, authors find out that additional tube implantation together with ectropion surgery did not contribute to better results.